# Full-Closed-Loop Time-Domain Integrated Modeling Method of Optical Satellite Flywheel Micro-Vibration

Yang Yu [1,2], Xiaoxue Gong [3], Lei Zhang [1,2,3,*], Hongguang Jia [1,2,3] and Ming Xuan [1,2,3]

1   Changchun Institute of Optics, Fine Mechanics and Physics, Chinese Academy of Sciences,
    Changchun 130033, China; yuyang171@mails.ucas.edu.cn (Y.Y.); jiahongguang@charmingglobe.com (H.J.);
    xuanming@charmingglobe.com (M.X.)
2   University of Chinese Academy of Sciences, Beijing 100049, China
3   Chang Guang Satellite Technology Co., Ltd., Changchun 130000, China; gxx@mails.ustc.edu.cn
*   Correspondence: zhanglei@charmingglobe.com; Tel.: +86-178-4334-4789

**Abstract:** Due to the micro-vibration of flywheels, the imaging quality of a high-resolution optical remote sensing satellite will be deteriorated, and the micro-vibration effect on the payload is complicated, so it is essential to establish a reasonable and accurate theoretical simulation model for it. This paper presents a method of full-closed-loop time-domain integrated modeling to estimate the impacts of micro-vibration generated by flywheels on optical satellites. The method consists of three parts. First, according to the satellites' micro-vibration influence mechanism in orbit, this paper establishes a full-closed-loop model framework. The overall model input is the instructions received and the output is the image shift. Second, in order to meet the requirements of time-domain simulation, this paper proposes a time-domain vibration source subsystem model in the form of cosine harmonic superposition, and it integrates vibration source, structural, control, and optical subsystem models to create a full-closed-loop time-domain analysis model that can obtain the responses of micro-vibration in time and frequency domains. Lastly, the author designs a ground experiment and compares simulation results with experiment results. Compared with the ground experiment, frequency error is less than 0.4% at typical responses. Although the amplitude error is large at some typical responses, the mean root square error is less than 35%. Based on the data, the proposed integrated modeling method can be considered as an accurate methodology to predict the impacts of micro-vibration.

**Keywords:** spacecraft micro-vibration; integrated modeling method; micro-vibration source model; full-closed-loop time-domain model; flexible satellite

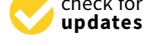

## 1. Introduction

The resolution of optical remote sensing satellites has now mostly reached the sub-meter level. With the increase in resolution level, more requirements have been posted on the stable mechanical environment of satellites. One of the main factors affecting the stability of satellites is the micro-vibration of flywheels. The flywheel is the actuator of the control subsystem; due to its small manufacturing errors and defects, it will produce small amplitude and wide-band disturbance forces or torques, which will lead to an instability of sensitive components. While this small vibration will not cause structural damage, it will significantly affect imaging resolution [1].

It is particularly significant to establish an accurate analysis model for research on the micro-vibration of high-resolution optical remote sensing satellites flywheels. The influencing characteristics of micro-vibration are the results of the interactions of vibration source, structural, control, and optical subsystems. Since the interaction characteristics are complex, it is difficult to reflect its true responses if models only consider one subsystem. At present, an integrated modeling method is widely adopted for micro-vibration analysis, which uses the tools of different disciplines to establish an integrated micro-vibration

analysis mathematical model. The model includes various subsystem models and their interaction processes, and it finally obtains the transfer characteristics of micro-vibration [2,3]. In early 1990s, Jet Propulsion Laboratory (JPL) was the first to adopt an integrated modeling that mainly integrated optical, control, structural and vibration source subsystems and proved the effectiveness of the method [4]. D.W. Miller et al. of the Space Systems Laboratory (SSL) of the Massachusetts Institute of Technology (MIT), USA, established a more detailed integrated modeling work based on the theory of linear time-invariant systems for Space Interferometry Mission (SIM), and they developed an integrated analysis software—DOCS (Dynamics Optics Controls Structures). DOCS integrated various subsystems and employed the Micro-Precision Interferometer Testbed (MPI) developed by S.S. Joshi et al. for SIM to obtain experimental data. By comparing measured and predicted closed-loop transfer functions, the reliability of the methodology was verified [5–8]. T.T. Hyde et al. from NASA developed the Integrated Modeling Environment (IME) for the James Webb Space Telescope (JWST). IME is able to perform micro-vibration analysis and structural–thermal–optical analysis to obtain dynamic wave front error and pointing error; another optical analysis software is also programed to obtain indicators of micro-vibration affecting the imaging performance of an optical subsystem according to the movement of optical components [9–11]. The two integrated software, IME and DOCS, fully considered the transmission mechanism of micro-vibration in each subsystem, effectively integrated the subsystem models, and developed a more various assessment of micro-vibration disturbance impact on optical indicators. The overall model input of the two kinds of software is micro-vibration disturbance source in the form of power spectral density (PSD), which has the advantages of simple calculation and wide coverage of frequency, but researchers did not analyze time-domain response characteristics. Pang et al. conducted an in-depth investigation and analysis on micro-vibration integrated modeling technology, and they drafted an overall framework of an integrated analysis system [12,13]. Zou et al. and Ge et al. proposed a time-domain analysis method based on time-domain calculations, which can better distinguish the effects of rigid-body low-frequency drift and elastic transient deformation, thereby contributing to obtain a steady time-domain response [14,15]. However, the real in-orbit control strategy of the control subsystem was not fully considered; the control subsystem was approximated as a second-order filter. Additionally, the harmonic superposition form of fixed frequency and amplitude is used as input, and the different characteristics of disturbance source output under different flywheel rotation speeds are not considered. Gong et al. used the Rosenblueth method and Fourier transform to transform the data between time-domain and frequency-domain and proposed a calculation method to convert frequency-domain image shift into a modulation transfer function (MTF) [16–18]. This method theoretically verified the influence of micro-vibration on imaging indicators, and it more conveniently reflected the influence of micro-vibration on imaging quality, but it did not consider the impact of the control subsystem. Chen et al. analyzed the causes of flywheel disturbances in detail from two aspects of structural disturbance and harmonic disturbance; the mechanical model for each disturbance is established corresponding to its characteristics. A flywheel disturbance test platform is established based on the Kistler table system. Two different types of flywheels are tested, and the dynamic characteristics of the flywheel disturbances are analyzed [19,20]. Li et al. researched the influence of flywheel disturbance on time-delayed and integration charge-coupled device (TDICCD) camera and satellite platform, which mainly includes two aspects: directly causing image shift and directly causing the jitter of a satellite platform. The influence of platform micro vibration on the image motion of an optical load is discussed from three aspects: the principle of the TDICCD camera, the classification of image motion, and the mechanism of image motion change caused by micro-vibration [21,22]. Alkomy et al. introduced a nonlinear dynamics model for these micro-vibrations with five degrees of freedom. The main advantage of the proposed model is that it analytically considers various disturbance sources [23]. Alcorn et al. presented a first-principles-based derivation of the motion equations for a spacecraft with react wheels subject to general static and dynamic imbalances. The resulting

formulation retains the true physics governing this fully coupled jitter phenomenon. As a result, energy and momentum checks are available by using this model [24]. Addari et al. investigated the coupled micro-vibration dynamics of a cantilever configured reaction wheel assembly mounted on either a stiff or flexible platform [25].

Compared with the disadvantages in the previous literature, we have improved the integrated modeling method. Based on a high-resolution optical remote sensing satellite, this paper proposes an integrated modeling method for flywheel micro-vibration problems. The novelty or contribution of this paper are as follows:

1.  The model fully considers the variable speed operation of flywheels caused by a control subsystem during the imaging process in orbit, and it realizes the coupling and integration of vibration source, structure, control, and optical subsystem with a high degree of realism and accuracy. A comparison of simulation results and experimental results shows that the frequency errors at typical responses are all less than 0.4%; although the amplitude errors at typical responses are larger, the mean root square error is less than 35%. It can be considered that the proposed simulation model can accurately predict the impacts of micro-vibration of flywheels in orbit;

2.  A new form of micro-vibration cosine harmonic superposition vibration source model is established, which not only satisfies the simulation of the overall model in the time-domain, but also considers the fact that a flywheel has different micro-vibration disturbance characteristics due to different flywheel rotation speed; the proposed model in this paper takes imaging mission instructions as input, and the vibration source subsystem is completely closed-loop within the overall frame;

3.  Future work will focus on inaccurate mode frequency and mode damping, the accuracy of analysis can be further improved by conducting satellite mode experiments and a finite element model check. The integrated modeling method proposed in this paper has broad application prospects, strong applicability to optical remote sensing satellites, and good engineering and practicality. It can be used to simulate and evaluate micro-vibration problems under more complex imaging modes in the future, and it can provide a reliable reference for the design of satellites micro-vibration vibration isolation and suppression.

Section 2 introduces the integrated modeling method and the modeling method of each subsystem. Section 3 shows simulation results when the imaging mode is set to adjust the attitude and control stable push-broom imaging. Section 4 presents a ground verification experiment, this section elaborates on the experimental conditions and procedures and compares the results. Sections 5 and 6 are the discussion and the conclusion of the research.

## 2. Integrated Modeling

### 2.1. Framework of Integrated Model

When satellites are in orbit, the control subsystem controls the rotation speed of flywheels according to the instructions received from the ground to generate control torques to adjust the attitude and maintain pointing accuracy. Micro-vibration disturbance generated by flywheels will be transmitted through satellite structural subsystem. When the frequency of micro-vibration disturbance is close to the mode frequency of optical subsystem elements, it will cause elements to jitter, thereby reduce imaging quality.

Based on the actual characteristics of a satellite, this paper establishes an analysis model including four subsystems of vibration source, structural, control, and optical subsystem; then, it creates a time-domain integrated model, the framework block diagram of which is shown in Figure 1. The vibration source subsystem can obtain the disturbance characteristic of flywheel micro-vibration, the structural subsystem has a transfer function in the micro-vibration physical space, the control subsystem is used to simulate various imaging modes, and the optical subsystem is used to convert the response of physical coordinates into imaging indicators. The reason why the model proposed in this paper is a full-closed-loop model is that the micro-vibration source subsystem of the closed-loop framework in the previous model is a type of external disturbance input, which means

that micro-vibration disturbance is the overall input. However, the proposed model in this paper takes imaging mission instructions as input, and the vibration source subsystem is completely closed-loop within the overall frame. Through such a method, the overall model will be free from external input disturbances and form a full-closed-loop.

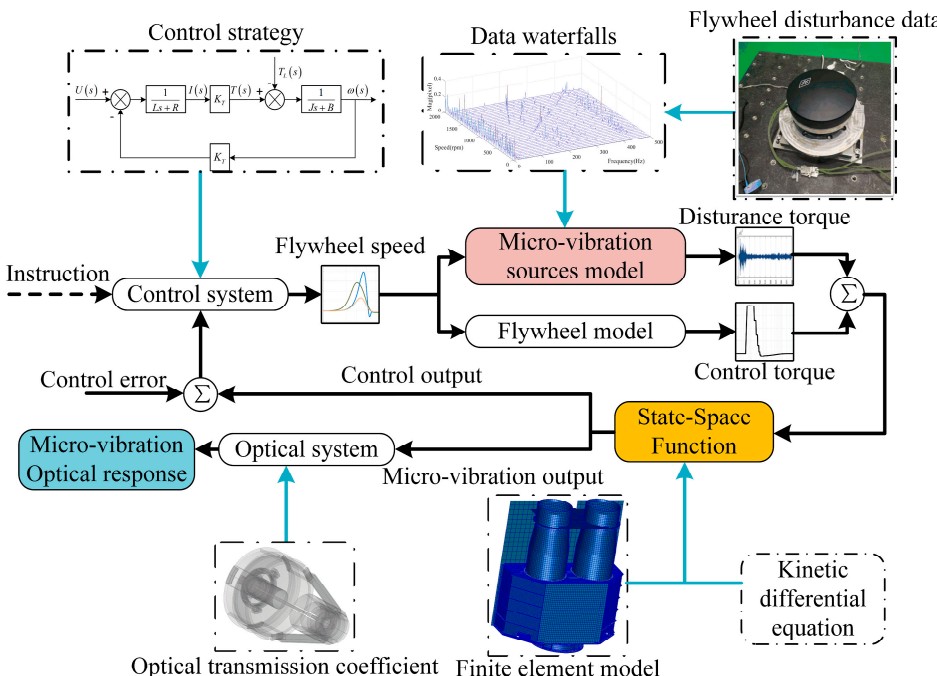

**Figure 1.** Framework block diagram of an integrated model.

The model integrates each subsystem model by connecting the input and output interfaces of each subsystem.

For the vibration source subsystem, this paper relies on the measured data to establish a model that takes flywheel rotation speed (time-domain) $n$ as input and obtains micro-vibration time-domain disturbance $T_d$ through the harmonic interpolation method, so that overall model can be simulated on time-domain.

For the structural subsystem, the authors built a state-space equation based on the differential equations of kinematics and satellite mode shape information. Its inputs include micro-vibration disturbance $T_d$ and control torque $T_c$, while the outputs are micro-vibration physical space response $y_{jit}$ and control output $y_{att}$.

For the control subsystem, the actual in-orbit control strategy is adopted, control output $y_{att}$ of the structural subsystem is taken as input, control torque $T_c$ is taken as output, and flywheel rotation speed $n$ is output according to the requirements of the micro-vibration disturbance source subsystem.

For the optical subsystem, the relationship between the micro-vibration physical space response $y_{jit}$ and imaging image shift $\Omega$ can be obtained according to the satellite optical load structure and the processing method based on optical theory.

## 2.2. Vibration Source Subsystem

In previous integrated modeling research, the models of the vibration source subsystem mainly applied two forms: (1) power spectral density and (2) harmonic superposition of fixed frequency and amplitude.

The first modeling method is to multiply the power spectral density function of the flywheel vibration source obtained from field measurements or theoretical analysis by the transfer function from the vibration source to the relevant position of interest, so as to obtain the frequency domain distribution function and ultimately obtain the micro-

vibration response information through integrators. However, it is easy to cause the loss of narrow-band characteristics under small damping during the process of integrators.

The second modeling method ignores the disturbing characteristics of the flywheel, that is, under an actual control subsystem, it has different rotation speeds during each imaging, resulting in different harmonic characteristics. Li has proven that micro-vibration causes the blurring of satellite imaging mainly due to the harmonic characteristics of the flywheel micro-vibration disturbance source [26]. So, this paper establishes a micro-vibration cosine harmonic superposition vibration source model, which not only satisfies the simulation of the overall model in the time-domain but also considers the fact that the flywheel has different micro-vibration disturbance characteristics due to the different flywheel rotation speeds.

The flywheel micro-vibration disturbance model based on experience is composed of discrete cosine harmonics, and its amplitude is proportional to the square of the flywheel rotation speed. The expression of disturbance force or disturbance torque $X_j(t)$ is:

$$X_j(t) = \sum_{i=1}^{n} C_{ij} n^2 \cos(h_{ij}t + p_j) \tag{1}$$

where $C_{ij}$ is the harmonic amplitude coefficient, $n$ is the flywheel rotation speed, $p_j$ is the initial phase, and $h_{ij}$ is the corresponding harmonic factor. Perform Fourier transform on Equation (1) and the ignore negative frequency interval, and get:

$$X_j(\omega) = \sum_{i=1}^{n} \pi C_{ij} n^2 [\cos(p_j) + j\sin(p_j)]\delta(\omega - h_{ij}n). \tag{2}$$

It can be concluded from Equation (2) that in a Campbell diagram with the flywheel rotation speed as the X axis, frequency as the Y axis, and amplitude as the Z axis, the peaks are at $\omega = h_{ij} * n$, and the peaks values are proportional to the square of the flywheel rotation speed. Typical disturbance peaks appear on a series of harmonic lines whose frequency is proportional to the flywheel rotation speed.

The satellite has three flywheels installed in three directions. A single flywheel includes the disturbance force (X, Y, Z-direction) and the disturbance moment (X, Y-direction). The disturbance moment in the Z direction is very small, so it is not considered. Table 1 shows the number of typical harmonics for each disturbance force/torque. As an example, a Campbell diagram of the disturbance torque (X-direction of flywheel) for one flywheel (Y-direction of satellite) and highlighted four harmonics are given in Figure 2, and Table 2 shows the relevant parameters of these four harmonics. When the flywheel rotation speed is $n$, the time-domain micro-vibration disturbance of one flywheel on the j-direction can be obtained by Equation (3):

$$T_d^j(t) = \sum_{i=1}^{n} A_{ij}(t) \cos[n(t) \cdot h_{ij} \cdot \tau_t(t) + p_{ij}(t)] \tag{3}$$

where $A_{ij}$ and $p_{ij}$ are respectively the amplitude and the initial phase of each order harmonic obtained by interpolating the harmonic characteristics in test data, and $\tau_t$ is the simulation time.

**Table 1.** Number of typical harmonics of each disturbance force/torque.

| Object Name | Number of Typical Harmonics |
|---|---|
| disturbance force X-direction | 74 |
| disturbance force Y-direction | 63 |
| disturbance force Z-direction | 96 |
| disturbance torque X-direction | 78 |
| disturbance torque Y-direction | 78 |

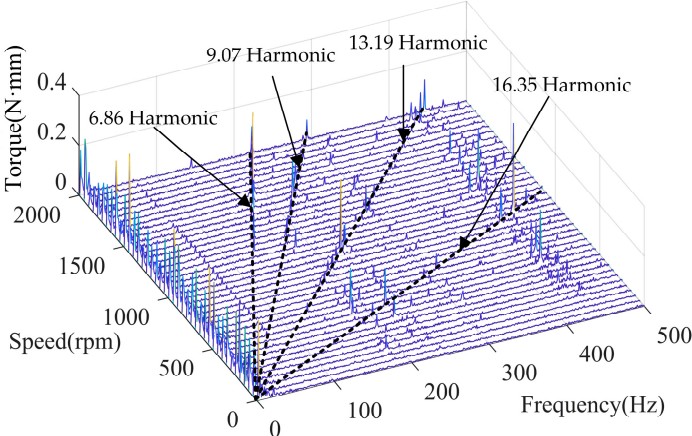

**Figure 2.** Campbell diagram of disturbance torque.

**Table 2.** Harmonic-related parameter.

| Harmonic Factor | $C_{ij}$ | $p_j$ | At −817.5 rpm | | At 974.4 rpm | | At 1136.5 rpm | |
|---|---|---|---|---|---|---|---|---|
| | | | Frequency (Hz) | Amplitude (N·mm) | Frequency (Hz) | Amplitude (N·mm) | Frequency (Hz) | Amplitude (N·mm) |
| 6.86 | $2.61 \times 10^{-7}$ | 2.43 | 93.47 | $-1.54 \times 10^{-4}$ | 111.41 | $2.05 \times 10^{-4}$ | 129.94 | $3.08 \times 10^{-4}$ |
| 9.07 | $2.49 \times 10^{-7}$ | −1.20 | 123.58 | $-3.03 \times 10^{-4}$ | 147.40 | $1.03 \times 10^{-4}$ | 171.80 | $1.16 \times 10^{-4}$ |
| 13.19 | $1.26 \times 10^{-7}$ | −1.44 | 179.71 | $-6.18 \times 10^{-5}$ | 214.21 | $1.05 \times 10^{-4}$ | 249.84 | $1.64 \times 10^{-4}$ |
| 16.35 | $2.90 \times 10^{-7}$ | 0.64 | 222.77 | $-1.40 \times 10^{-4}$ | 265.52 | $2.35 \times 10^{-4}$ | 309.70 | $3.91 \times 10^{-4}$ |

### 2.3. Structural Subsystem

The structural subsystem starts with physical coordinates in second-order form in the time-domain. By transforming the differential equation into a state-space equation, the physical coordinates convert to mode coordinates, and the output is transformed into a first-order form so that it can be linked to other subsystems. The advantages of this approach are as follows. As a complex system with multiple degrees of freedom, a satellite is a Multiple-Input Multiple-Output (MIMO) system. A state-space equation can be simplified by the mode selection matrix and becomes a Single-Input Single-Output (SISO) system, and because it is transformed based on the dynamic differential equation, it satisfies observability and controllability. Establish satellite dynamic differential Equation (4):

$$M\ddot{x} + C\dot{x} + Kx = T_d + T_c \tag{4}$$

where $M$ is the mass matrix, $C$ is the damping matrix, $K$ is the stiffness matrix, and $x$ is the displacement of nodes in physical coordinates. Based on the finite element model and using the mode superposition method, Equation (4) is transformed from physical coordinates to mode coordinates form:

$$\ddot{q} + 2\xi\omega\dot{q} + \omega^2 q = \varphi^T (T_d + T_c) \tag{5}$$

where $q$ is the mode coordinates, the first $r$ states are taken (the three translational rigid-body modes are not considered), $\xi$ is the diagonal mode damping matrix, $\omega$ is the diagonal frequency matrix, and $\varphi^T$ is the mode shape matrix normalized by the mass matrix. Equation (6) is Equation (5) described in a state-space equation.

$$\begin{bmatrix} \dot{\eta} \\ y_{att} \\ y_{jit} \end{bmatrix} = \begin{bmatrix} A & B_{T_c} & B_{T_d} \\ C_{att} & D_{T_d\_att} & D_{T_c\_att} \\ C_{jit} & D_{T_d\_jit} & D_{T_c\_jit} \end{bmatrix} \begin{bmatrix} \eta \\ u_{T_d} \\ u_{T_c} \end{bmatrix} \tag{6}$$

where the state vector $\eta = \begin{bmatrix} \dot{q} & q \end{bmatrix}^T$, $y_{att}$ and $y_{jit}$ are the control output and the micro-vibration response output, respectively, $u_{T_d}$ and $u_{T_c}$ are the control input and the micro-vibration disturbance input, respectively. The system matrix expression is shown in Equation (7),

$$
\begin{bmatrix} A & B_{T_c} & B_{T_d} \\ C_{att} & D_{T_d\_att} & D_{T_c\_att} \\ C_{jit} & D_{T_d\_jit} & D_{T_c\_jit} \end{bmatrix} = \begin{bmatrix} -\omega^2 & -2\xi\omega & \phi^T\alpha_{T_d} & \phi^T\alpha_{T_c} \\ 0 & I & 0 & 0 \\ 0 & \phi\beta_{T_d} & 0 & 0 \\ 0 & \phi\beta_{T_d} & 0 & 0 \end{bmatrix} \tag{7}
$$

where $\alpha$ and $\beta$ are the mode selection matrices. The transmission characteristics of satellite structural dynamics in the time-domain can be obtained by Equation (6).

To achieve mode coordinates $q$, a finite element model, as shown in Figure 3, is built in finite element software (Nastran). The mode shape is shown in Figure 4. The satellite model has a total of 234,570 units and 330,118 nodes. In this model, flywheels are approximated as three mass nodes. By adjusting mode selection matrix $\alpha$, micro-vibration disturbance input Td and control input Tc are applied to the mass nodes of flywheels. Take the natural mode within 0–500 Hz (a total of 232 states). Adjusting mode selection matrix $\beta$ can locate the output position at concerned nodes, including the primary mirror, secondary mirror, focal plane, and star sensor (as detailed in Sections 2.4 and 2.5). In the finite element model, most of the structures are solid elements with types of penta6 and hex8, and the thin-walled structures such as solar array use shell elements with types of tria3 and quad4. This meshing method meets the requirements of the aerospace field for finite element analysis. No loading is applied, and the constraint is a free boundary condition.

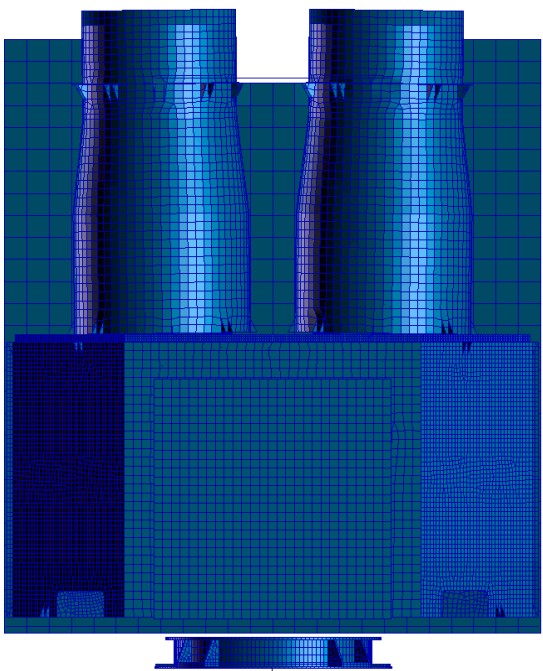

**Figure 3.** Finite element model.

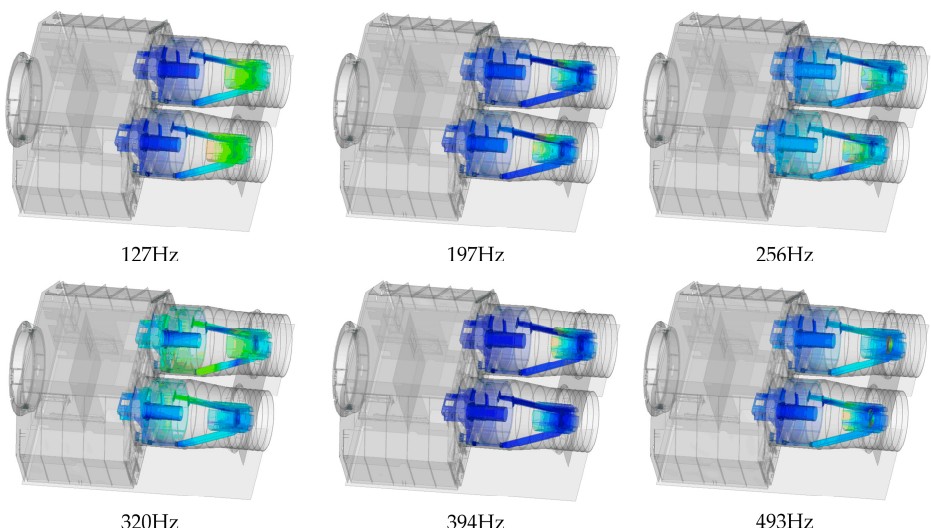

127Hz　　　　197Hz　　　　256Hz

320Hz　　　　394Hz　　　　493Hz

**Figure 4.** Typical mode shapes.

In the finite element model of the structural subsystem, the simulation results in the frequency domain often show many responses that are not present in the test or have a very small amplitude. The reason is that the uniform mode damping is adopted in the modeling, which causes the response amplitude to be inconsistent with the reality, or the finite element model is different from the actual structure of the satellite. Therefore, we conducted a mode test on the satellite by using the hammer excitation method, compared the frequency of the main modes of the satellite, modified and checked the mode damping of some important natural modes, and modified the mode damping in the state-space equation. Finally, the model after modified mode damping is used for simulation analysis. Table 3 shows the modified mode damping of some typical modes.

**Table 3.** Modified mode damping.

| Frequency (Hz) | Mode Damping (%) |
|:---:|:---:|
| 127.7 | 0.412 |
| 197.3 | 0.301 |
| 255.9 | 0.382 |
| 320.3 | 0.493 |
| 394.5 | 0.594 |
| 493.2 | 0.317 |
| other | 0.500 |

*2.4. Control Subsystem*

In this paper, the control subsystem model adopts the real attitude control strategy of the satellite (shown in Figure 5). When the control subsystem is running, imaging mission instructions are uploaded to the satellite from the ground remote system. The center control computer generates control instructions according to the proportional-derivative (PD) control strategy. According to the control instructions, flywheels generate control torque, which acts on the structural subsystem to change the attitude. The star sensor measures attitude information and transmits it to the central control computer for calculating the next control instructions, thus forming a closed-loop attitude control subsystem.

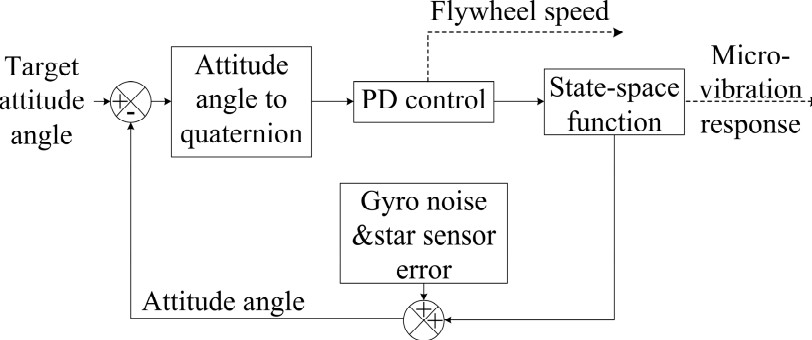

**Figure 5.** Block diagram of the control subsystem.

A traditional attitude control subsystem regards a satellite as a rigid body and obtains the attitude information after control torque is applied through a Euler dynamic equation. However, in this paper, attitude information is obtained through a state-space equation (which regards a satellite as a flexible body). Since the star sensor is the component that obtains attitude information in orbit, the response information of the finite element node of the star sensor is selected as the input of the control subsystem in Section 2.3. The needs of attitude adjustment and stability control would be met by the feedback control process.

*2.5. Optical Subsystem*

The decrease in the imaging resolution of optical remote sensing satellites is the ultimate effect of micro-vibration, so it is more intuitive to estimate the impact of micro-vibration by taking the imaging resolution index as the final output result. Micro-vibration will cause the optical element to jitter and deviate from its initial position. When the position of any optical element changes, the position of the image point also changes, resulting in a decrease in resolution.

The optical model used for micro-vibration analyses can be described as a Taylor expansion of the optical path lengths of a uniformly incident beam through the optical subsystem. Since the magnitude of micro-vibration disturbance is very small, high-order terms can be ignored. Image point movement can be decomposed as a function of the displacement and rotation angle of each optical element:

$$\Omega = \sum_{i=1}^{N} \frac{\partial L}{\partial T_i} \Delta T_i + \sum_{i=1}^{N} \frac{\partial L}{\partial R_i} \Delta R_i \tag{8}$$

where $\Delta T_i$ and $\Delta R_i$ are the displacement and rotation angle vectors of each optical component, and $\frac{\partial L}{\partial T_i}$ and $\frac{\partial L}{\partial R_i}$ are the displacement and rotation angle transfer coefficients of each optical component, which can be obtained by optical calculation software. The response $y_{jit}$ caused by an optical element in physical coordinates can be converted into image point movement $\Omega$ by Equation (8).

The optical subsystem in a satellite is a Cassegrain coaxial two-mirror optical system, and the corresponding schematic diagram is given in Figure 6. The optical subsystem is mainly composed of two reflecting mirrors and one focal plane. For such an optical system, micro-vibration affects imaging mainly by the micro-vibration response of the primary mirror, secondary mirror and focal plane. Therefore, in Section 2.3, the three nodes' information is selected as a micro-vibration response output.

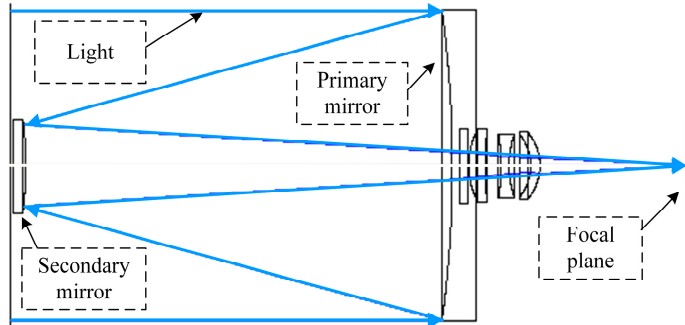

**Figure 6.** Schematic diagram of optical subsystem.

## 3. Simulation and Calculation

By changing the instructions parameters of the control subsystem, the integrated modeling method can predict the impact of micro-vibration under a variety of imaging modes, including push-broom, gaze, large side swing, and others. In this paper, the simulation imaging condition is push-broom imaging after adjusting attitude, which is widely employed in optical imaging satellites. The relevant parameters set for the proposed integrated model are shown in Table 4.

**Table 4.** Simulation instructions parameters.

| Object Name | Simulation Parameter |
|---|---|
| orbit height | 500 km |
| initial attitude angle | [0 0 0]° |
| target attitude angle | [1 2 3]° |
| flywheel's moment of inertia | 0.00321 kg·m² |
| star sensor error | [5 5 10]″ |
| gyro constant drift | 0.0008°/s |

As an example of simulation results, the response curve of the XYZ-direction rotation angle and the Y-direction displacement of the secondary mirror node and flywheel speed are shown in Figures 7 and 8.

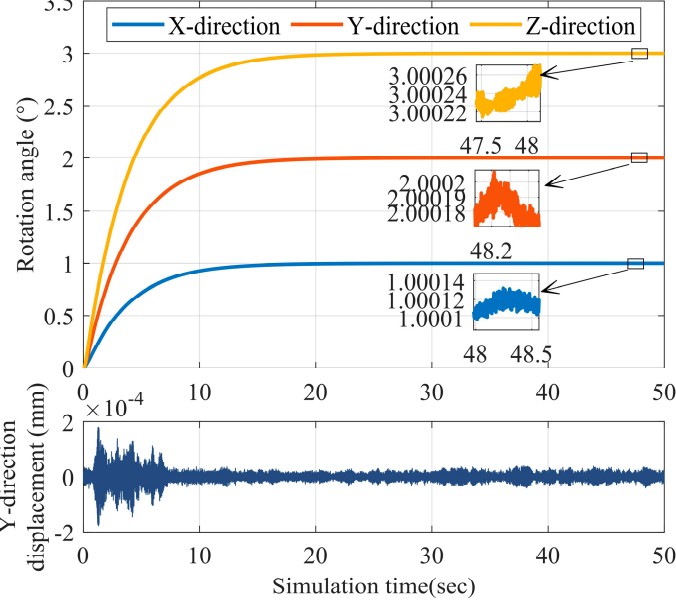

**Figure 7.** Time-domain response of secondary mirror node.

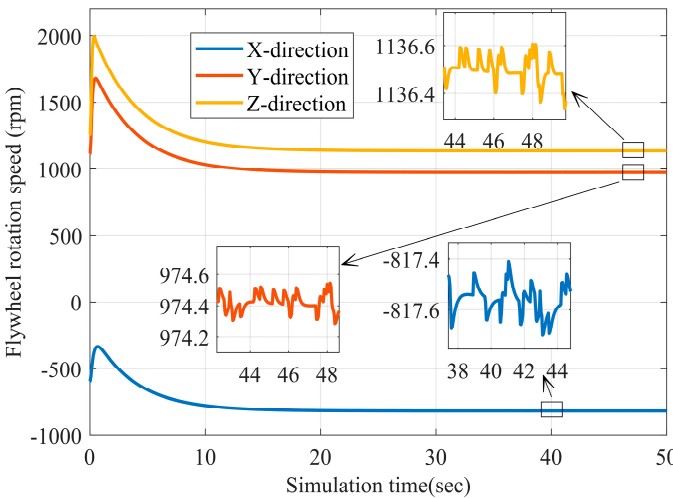

**Figure 8.** Flywheel rotation speed curve.

The satellite adjusts its attitude angle from the initial attitude angle to the target attitude angle, and after about 30 s, it is stable and turns into the imaging stage. The focus of this paper is the micro-vibration response during imaging in the last 20 s. The response of micro-vibration could be observed when curves were amplified. Flywheel rotation speeds in the stage of push-broom imaging are not at a fixed value due to the addition of star sensor error, gyro constant drift, and micro-vibration feedback. Figure 9 shows the movement of image points in the XY-plane. Figure 10 shows the frequency domain response diagram of the Y-direction which is charge coupled device camera (CCD) push-broom direction.

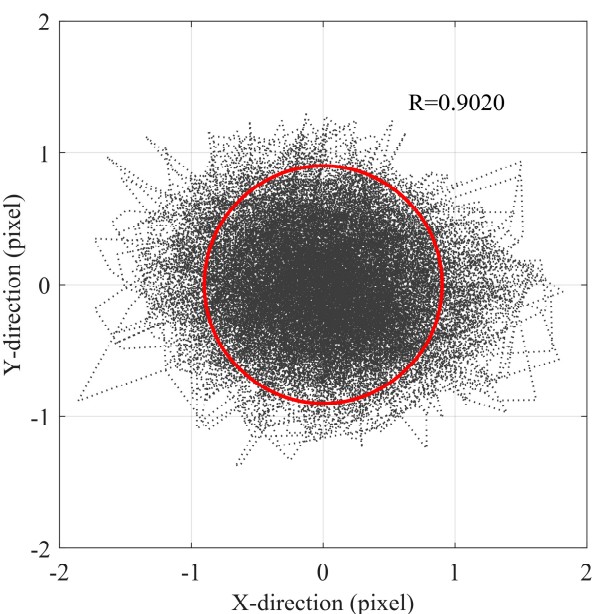

**Figure 9.** The movement of image point in the XY-plane.

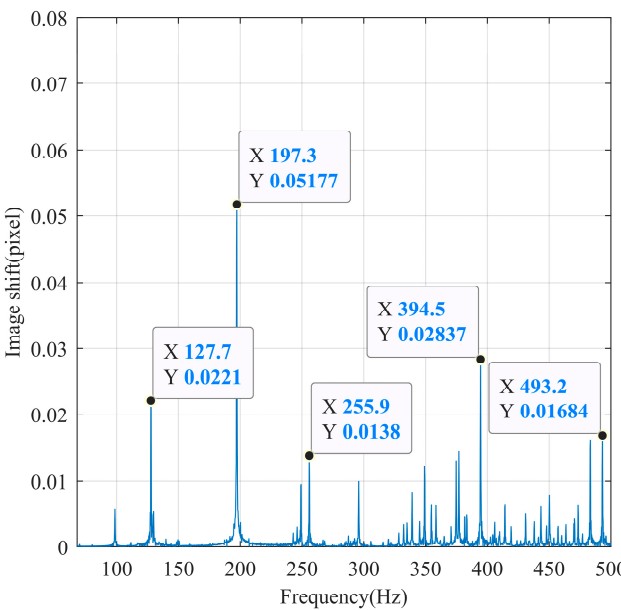

**Figure 10.** Frequency-domain response of the simulation results.

## 4. Verification

### 4.1. Ground Experiment

Due to air resistance and the inability to simulate free boundary conditions, it is difficult to design a ground experiment to simulate the process of satellite attitude rotation, and the flywheel rotation speed change process will be quite different from the simulation result. Therefore, the flywheel control system has made the experimental flywheel rotation speed run according to the simulation curve value. A ground experiment platform built is shown in Figure 11. The experimental equipment includes a satellite prototype, collimator, integrating sphere, flywheel control system and imaging equipment.

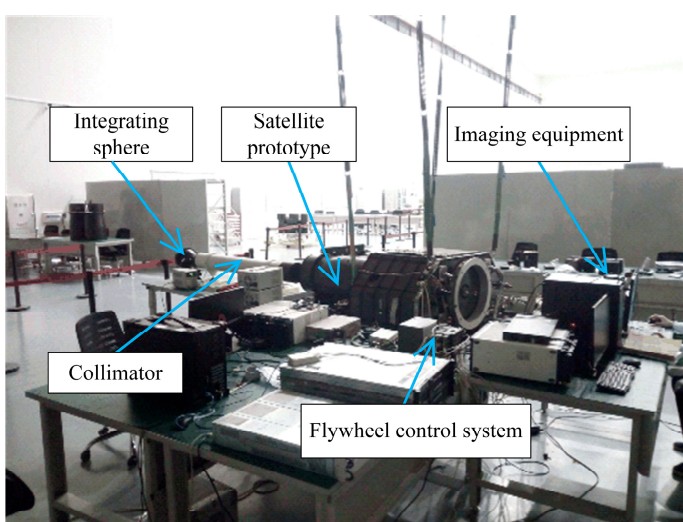

**Figure 11.** Micro-vibration ground experiment.

The integrating sphere is used as a light source, and a collimator is used to image the slit target to simulate the incidence of infinite objects. According to the focal length of the collimator and optical subsystem, the system aligns a line target into a magnified image, which is calculated according to the line target image scale of 5 pixels, and the slit width is 13.75 μm. When the width of the slit image is 5 pixels, the slit diffraction intensity is small, and the internal brightness is uniform to ensure the experimental accuracy (0.1 pixel). The

focal plane uses push-broom CCD, and the line transfer time is 100 microseconds, so the sampling frequency is 10,000 Hz. Figure 12 is the schematic diagram of the light path.

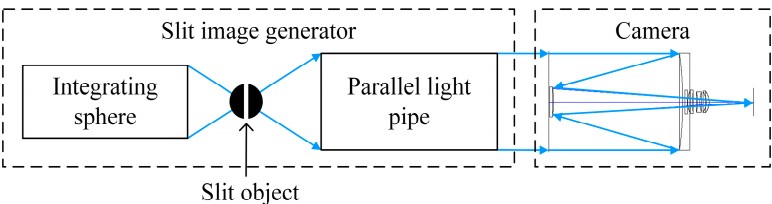

**Figure 12.** Schematic diagram of the light path.

### 4.2. Result Verification

After obtaining the target image, each line of the image is smoothed and filtered to eliminate the sudden changes in image brightness caused by noise (Figure 13).

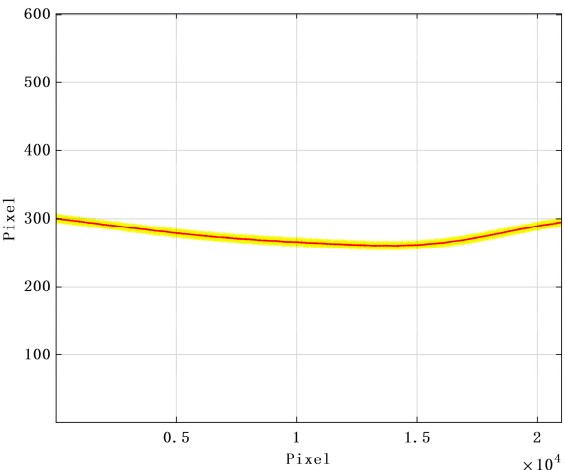

**Figure 13.** Target image and line brightness centroid position curve (red line).

The frequency-domain responses of ambient noise at 0 rpm and the experiment are shown in Figure 14. The frequency-domain responses of simulation and the experiment are shown in Figure 15. Table 5 shows the comparison of simulation and experimental results of root mean square (RMS), frequency, and amplitude at typical response.

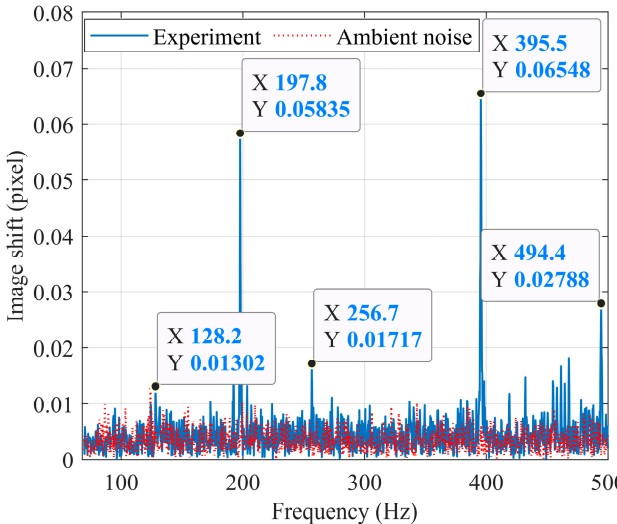

**Figure 14.** Frequency-domain response of ambient noise and experiment.

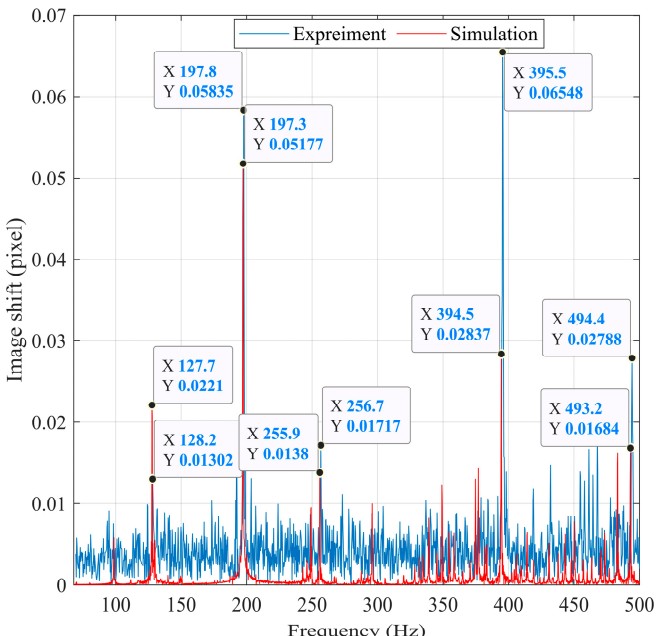

**Figure 15.** Frequency-domain response of simulation and experiment.

**Table 5.** Comparison of RMS and amplitude/frequency at typical response.

| | RMS | | |
|---|---|---|---|
| | **Simulation Results (Pixel)** | **Experimental Results (Pixel)** | **Errors** |
| | 0.4391 | 0.3275 | 34.08% |
| | **Frequency** | | |
| | **Simulation Results (Hz)** | **Experimental Results (Hz)** | **Errors** |
| response 1 | 127.7 | 128.2 | 0.39% |
| response 2 | 197.3 | 197.8 | 0.25% |
| response 3 | 255.9 | 256.7 | 0.31% |
| response 4 | 394.5 | 395.5 | 0.25% |
| response 5 | 493.2 | 494.4 | 0.24% |
| | **Amplitude** | | |
| | **Simulation Results (Pixel)** | **Experimental Results (Pixel)** | **Errors** |
| response 1 | 0.022 | 0.013 | −69.23% |
| response 2 | 0.052 | 0.058 | 10.34% |
| response 3 | 0.014 | 0.017 | 17.64% |
| response 4 | 0.028 | 0.065 | 56.92% |
| response 5 | 0.017 | 0.028 | 39.29% |

## 5. Discussion

According to Table 5, the conclusions are as follows:

1. The maximum error of the typical response frequency value has a relatively small value of 0.39%. It proves that the proposed integrated modeling method can accurately capture the typical characteristic response of flywheel micro-vibration disturbance, and it has important reference significance for satellite micro-vibration isolation design.

2. Figure 16 shows the time-domain data of simulation results and experimental results of image shift in the Y-direction. It is shown that the results are close to the same magnitude in the time-domain. The RMS of the simulation results is 0.4391 pixels, and the RMS of the experimental results is 0.3275 pixels, with an error of 34.08%. According to the results of Liu et al. from modeling and analysis of Solar Dynamics Observatory (SDO) micro-vibration, the RMS predicted by the simulation results is usually 1.5–2 times that of

the experimental results [27]. The error of RMS in this paper is 1.34 times, which is within a reasonable range.

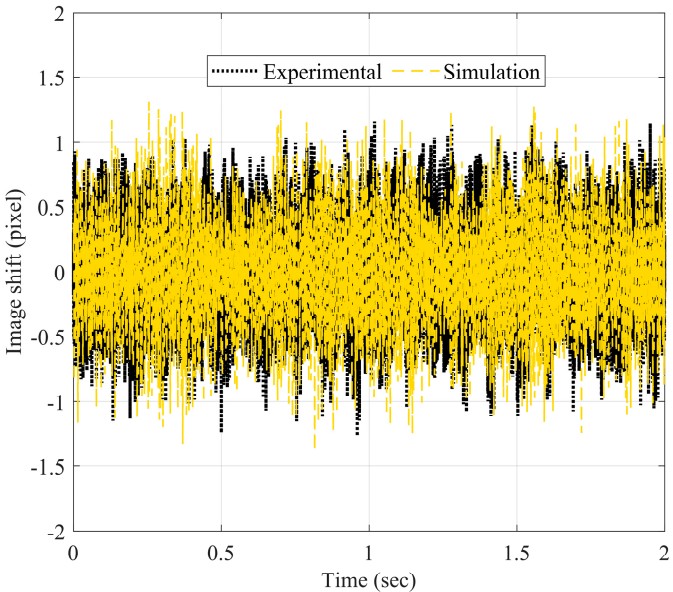

**Figure 16.** Comparison of time-domain results.

3. The amplitude results of typical responses are relatively different in terms of errors range, due to/stemming from errors inherited in experimental results and inaccurate mode damping values. The transfer function derived from the Laplace transform of the dynamic differential Equation (4) is as follows

$$H_{ij}(\omega) = \sum_{r=1}^{N} \frac{\phi_{ri}\phi_{rj}}{m_r[(\omega_r^2 - \omega^2) + j2\xi_r\omega_r\omega]} \tag{9}$$

where the value of mode damping $\xi_r$ of each mode determines the response amplitude. Although the relevant mode damping has not been precisely adjusted, the accuracy of RMS analysis results is still better than that of previous research results. After the mode damping is adjusted, an analysis error can be further reduced, and analysis accuracy can be further improved.

4. In the finite element simulation, we found that typical modes of sensitive optics in satellites include the response of the secondary mirror at 127, 394, and 493 Hz and the response of the primary mirror at 197, 256, and 320 Hz. From the simulation results (Figure 4), due to the micro-vibration disturbance of the flywheel, the satellite natural modes are excited at 127, 197, 256, 394, and 493 Hz. However, since there is no disturbance input near 320 Hz, the natural mode of the main mirror does not respond at 320 Hz, so there is no peak at 320 Hz in the frequency domain results of the imaging pixels. In addition, there are some inherent modes with small amplitude (less than 0.01 pixel) in Figure 10, whose magnitude is at the same magnitude as the experimental environment noise, and will not have a significant impact on satellite imaging.

## 6. Conclusions

The integrated modeling work presented in this paper pays more focus on micro-vibration response characteristics that are caused by continuous controlled flywheel in time-domain. Similar to previous research studies, this paper integrates vibration source, structural, control, and optical subsystem models. However, the integrated model applies the instructions of adjusting attitude as input, while output is the information of optical performance index in time and frequency domains. Comparing with previous integrated modeling methods, the method in this paper can better simulate a satellite's status during

in-orbit operation, and output real-time predictions of the impacts of flywheel micro-vibration on imaging under different imaging modes.

**Author Contributions:** Conceptualization, Y.Y. and X.G.; methodology, Y.Y.; software, Y.Y.; validation, Y.Y.; formal analysis, Y.Y.; investigation, Y.Y.; resources, Y.Y.; data curation, Y.Y.; writing—original draft preparation, Y.Y.; writing—review and editing, Y.Y. and X.G.; visualization, L.Z.; supervision, H.J.; project administration, M.X.; funding acquisition, L.Z. All authors have read and agreed to the published version of the manuscript.

**Funding:** This research was funded by National key Research and Development plan of China, No.2016YFB0500904.

**Institutional Review Board Statement:** Not applicable.

**Informed Consent Statement:** Not applicable.

**Data Availability Statement:** Not applicable.

**Conflicts of Interest:** The authors declare no conflict of interest.

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
