# Peer review of "Full-Closed-Loop Time-Domain Integrated Modeling Method of Optical Satellite Flywheel Micro-Vibration"

_applsci, doi:10.3390/app11031328_

Round 1

Reviewer 1 Report

The paper proposes a model of the micro-vibration generated by attitude control flywheels and their impact on the optics mounted on a satellite.

The topic is of interest and the results are satisfactory, even if the paper lacks clarity in some sections and many details are left without a clear explanation. The reviewer feels that the paper may become worth a Journal publication, but in order to achieve this goal, an important revision is needed, both from the point of view of writing clarity and content improvement.

Main aspects to be clarified:

  • Authors claim that one of the main novelties of this paper is the introduction of the complete modeling of the vibrations generated by the control flywheel, achieved through a functional description of the disturbance force/torque; the disturbance is indeed modeled as the sum of discrete cosine harmonics whose amplitude is proportional to the squared flywheel speed, and frequency proportional to the flywheel speed. Figure 2 highlights some of these components (harmonics), and from that graph, it can be seen that the order of those components is in the range between 6 and 24 (4 highlighted harmonics), but the reader does not have any idea on how to set the parameters contained in the description of the disturbance, nor he is given any indication of the values that have been used in the performed simulation. This should be a content of the revised paper, and all the details needed to reproduce the obtained results should be available throughout the paper.
  • Authors claim that one of the main advantages in fully describing the disturbances generated by the control wheel is due to the capability to describe that disturbance even when the flywheel speed is changing. This statement conflicts with what is declared on page 9: “The focus of this paper is the micro-vibration response during imaging in the last 20 seconds of the simulation.” During the last 20 seconds, the flywheel speed is almost constant as it can be seen in Figure 8, and this will generate fixed frequency disturbances, and therefore the importance of the suggested description is not felt by the reader as fundamental as claimed. In order to have a clear demonstration of its importance the suggestion is to run a simulation using a standard description of the disturbances and to compare the results with the already obtained ones.
  • The simulation response shows many frequency components that are not present in the real system. This behavior may be due to the damping coefficients that could be refined (as stated in the paper) but could be due to issues in the model identification process. More discussion should be dedicated to the meaning of each frequency component: many of the harmonics that are present in simulations and missing in the experiments, have amplitudes that are not negligible. The description of the full set of modes that may affect the optic behavior should be provided, at least in terms of involved frequencies.

Other minor aspects to be addressed:

  • Page 1 line 38: instead of “Due to” maybe “Since” would be better
  • Page 2 line 70: unclear sentence, please rephrase
  • Page 2 line 83: it seems there is a missing “are”
  • Page 3 lines 105-108: difficult to read, please rephrase
  • Page 3 line 111: it seems there is a missing “used”
  • Page 4 line 150: unclear sentence, please rephrase
  • Page 4 lines 164-168: the presented graph is usually referred to as Campbell diagram
  • Page 8 line 251: unclear sentence, please rephrase
  • Page 11 lines 315-316: unclear sentence, please rephrase
  • Figure 13: No axis on the figure
  • Page 12 lines 320-321: unclear sentence, please rephrase
  • Page 13 lines 328-331: the matching of the excited frequency is mainly due to the modal model to match the real frequencies of the system under study. This should be stated more clearly.
  • Page 14 lines 357-358: unclear sentence, please rephrase

Reviewer 2 Report

The paper addresses an issue of full-closed-loop time-domain integrated modeling to estimate the impacts of micro-vibration generated by flywheels on optical satellites. The aim of the paper is to an accurate methodology to predict the impacts of micro-vibration.

The work is based on theoretical researches and experimental verifications. The methodology of data collection and processing is well presented. The results interpretation and the conclusion are clear.

The paper is well written and expressed.

Reviewer 3 Report

This work “Full-closed-loop Time-domain Integrated Modeling Method of Optical Satellite Flywheel Micro-vibration. In this manuscript, a method of full-closed-loop time-domain integrated modeling to estimate the impacts of micro-vibration generated by flywheels on optical satellites is presented. Although, the work is interesting, some points should be improved:

The abstract of the article can be restructured. Mention the problem statement for the better understanding of readers at the beginning of abstract. Moreover, mention the name of software on which FEM simulations are carried out. There are few grammatical errors throughout the paper, which need to be corrected. Try to avoid unnecessary long sentences. Literature review is ambiguous; include some more recent state of the art papers in Literature review for better understanding. Try to add some recent work as well, as many of the papers are decade ago. The Introduction part of the article must be revised to make it better structured for the readers. Try to explain the previous work related to different aspects of the current research and connect it with the problem statement in the end i.e. identifying the gap and why was this model necessary to develop. An intense revision is required in this section. In the last paragraph of the introduction section, mention the novelty of this paper with previous state of the art research. Rather than mentioning the results/conclusions of the manuscript. Moreover, mention the applications of this work. Link the novelty with previous published work as well. Discuss the results more deeply and explain the physics behind it. For example, add explanation of Fig. 10. If possible, compare the numerical/experimental data with the data already present in literature for validation in graphical form. In Fig. 13, if possible, use white background. In line 203-210, mention the type of meshing, why that meshing is used, loading and boundary conditions in detail. Why are you using state space equation for altitude control information rather than using a Euler equation? Explain it with proper references. In line 281, if we use conventional model (not considering state-space equation) and we analyse time response. Won’t it be more sensitive and accurate?

Round 2

Reviewer 3 Report

The paper as bene deeply revised and for this reviewer it can be published.